# Sociodemographic Determinants in Breast Cancer Screening among Uninsured Women of West Texas

**DOI:** 10.3390/medicina58081010

**Published:** 2022-07-28

**Authors:** Brooke Jensen, Hafiz Khan, Rakhshanda Layeequr Rahman

**Affiliations:** 1School of Medicine, Texas Tech University Health Sciences Center, Lubbock, TX 79430, USA; brooke.jensen@ttuhsc.edu; 2Julia Jones Matthews Department of Public Health, Texas Tech University Health Sciences Centre, Lubbock, TX 79430, USA; hafiz.khan@ttuhsc.edu; 3Department of Surgery, School of Medicine, Texas Tech University Health Sciences Center, Lubbock, TX 79430, USA; 4Southwest Cancer Center, University Medical Center Lubbock, Lubbock, TX 79430, USA

**Keywords:** breast cancer, mammogram, medically underserved, BI-RADS

## Abstract

*Background and Objectives*: Early detection through appropriate screening is key to curing breast cancer. The Access to Breast Care for West Texas (ABC4WT) program offers no-cost mammography to underserved women in West Texas. The U.S. Preventative Task Force (USPSTF) guidelines are breast cancer screening guidelines which suggest screening for all women at the age of 50 years. The focus of this study was to identify sociodemographic barriers and determinants for breast cancer screenings, as well as screening outcomes, in low income, uninsured, or under-insured communities in West Texas. *Materials and Methods*: The ABC4WT program’s patient database was queried from 1 November, 2018, to 1 June, 2021, for sociodemographic variables, screening history, and results to identify high-risk groups for outreach. The American College of Radiology’s risk assessment and quality assurance tool, BI-RADS (Breast Imaging-Reporting and Data System), a widely accepted lexicon and reporting schema for breast imaging, was used for risk differentiation. *Results:* The cancer rate for ABC4WT’s program was significantly higher than the national mean (5.1), at 23.04 per 1000 mammograms. Of the 1519 mammograms performed, women between 40 and 49 years old represented the highest percentages of BI-RADS 4 and 5 (42.0% and 28.0%, respectively; *p* = 0.049). This age group also received 43.7% of biopsies performed and comprised 28.6% (*n* = 10) of cancers diagnosed (*n* = 35) (*p* = 0.031). Additionally, participants with a monthly household income of less than USD 800/month/person were more likely to result in a cancer diagnosis (70.6%) than higher incomes (29.4%) (*p* = 0.021). *Conclusions:* These determinants most starkly impacted women 40–49 years old who would not have been screened by U.S. Preventative Services Task Force (USPSTF) guidelines. This population with increased cancer risk should be encouraged to undergo screening for breast cancer via mammography.

## 1. Introduction

Breast cancer remains the leading cause of death amongst women in the United States, with estimations of 276,480 new cases and 42,170 deaths in the year 2020 [1]. According to reports from the Texas Department of State Health Services, of all female cancers, the age-adjusted breast cancer incidence rate in Texas during 2013–2018 was the highest amongst all races (113.9 per 100,000 women) with an average of 17,329 deaths per year [2,3]. Of the Texas women at risk, non-Hispanic White women had the highest incidence rate (126.4), which was closely followed by the rate for non-Hispanic Black women (121.2) and much larger than the rate for Hispanic women at (90.6 per 100,000 women) [2].

Mammogram screening has been recognized as a particularly useful tool in early disease detection, resulting in a better prognosis and diminished death rate from breast cancers [4]. Despite the associated benefits [5], the overall coverage of mammogram screenings in the United States is still very low. A plethora of factors contribute to this low coverage among indigent women who lack health insurance coverage for screening mammography [6,7], with higher odds of these women diagnosed with advanced stages of the disease [7,8,9]. Ethnicity, health, and socioeconomic disparities (e.g., poverty, cultural beliefs, and marital status) contribute to the low compliance with mammography screening and decreased access to prevention programs, thus increasing the disparity in breast cancer rates and outcomes amongst underserved women [10,11,12,13]. Additional factors that have been noted to decrease compliance with screening mammography are the fear of pain or of a poor screening outcome, educational attainment, geographical location, lack of awareness, and co-morbidities [10,11,12,13]. Studies have also shown that compliance with mammography screening varies greatly by age, educational level, access to a primary care physician, and insurance coverage status [14,15,16].

The Access to Breast Care for West Texas (ABC4WT) program, funded by the Cancer Prevention and Research Institute of Texas (CPRIT), was initiated in 2010 in the Texas Panhandle (Council of Government Region-1 (COG-1)). The program now extends to both the South Plains (COG-2) and West Central Texas (COG-7), with an overarching goal of reducing the rate of advanced breast cancer, as well as the economic burden on the state posed by the treatment of advanced diseases in the uninsured/underinsured population (Figure 1). The project was performed by the Texas Tech University Health Sciences Center’s Breast Center of Excellence (TTUHSC-BCE).

The three COGs covered by ABC4WT were predominantly (76–81%) inhabited by White individuals (both Hispanic and non-Hispanic individuals were included in this category because they were combined by the Census Bureau). Educational levels varied by region; however, when comparing the largest city of each COG, Amarillo (COG-1) had the lowest educational attainment, with only 5.1% of the population holding a bachelor’s degree or higher. Interestingly, the United States Census Bureau found that the largest cities in each COG (Amarillo (COG-1), Lubbock (COG-2), and Abilene (COG-7)) had similar population incomes, with the most frequently reported household income being USD 50,000 to USD 74,999. The economic burden of breast cancer in these regions was remarkable due to the increased incidence rates relative to the state’s average. In the year 2018, the age-adjusted breast cancer incidence rate (per 100,000 women) for COG-2 averaged 113.1, whereas COG-1 (115.2) and COG-7 (114.8) incidence rate averages were more similar to the state average rate of 115.2 [17]. The age-adjusted mortality rates (per 100,000 women) in COG-2 (23.1) and COG-7 (25.5) far exceeded the state average of 19.9; however, the mortality rate in COG-1 (15.4) was below the state average [17].

Although mammogram screening is an invaluable tool that can detect breast cancer in its early stages, the lack of compliance and access to these resources is a significant contributor to the high mortality and morbidity rate in this population. This study examined the role of socioeconomic and sociodemographic factors in the non-compliance of underserved/uninsured women treated by ABC4WT in three regions of West Texas (COG-1, COG-2, and COG-7).

## 2. Materials and Methods

The basis for this study was the comprehensive community outreach program, Access to Breast Care for West Texas (ABC4WT), funded by CPRIT. ABC4WT is an outreach program targeting the underserved (uninsured/under-insured) population of the Texas Panhandle, South Plains, and Central West Texas regions (Council of Government (COGs) 1, 2, and 7, respectively) (Figure 1). This project represents a public–private–community partnership to increase breast cancer screening in the target region. The focus of this study was to identify barriers and outcomes of breast cancer screening (provided at no cost to uninsured women), utilizing the American College of Radiology (ACR) guidelines that recommend annual mammograms for women aged 40 and above, or 10 years younger than the youngest first-degree relative with cancer [18]. Average-risk women under 40 years of age were offered diagnostic mammograms only if they were symptomatic as confirmed by clinical exam. All mammograms were performed at Mammography Quality Standards Act (MQSA)-certified facilities utilizing the ACR reporting lexicon [19,20].

The ABC4WT outreach program maintains a prospective database for all participants served by the project. The data were collected on sociodemographic variables, health insurance status, and prior screening history. The prospectively collected data, ranging between 1 November, 2018, and 1 June, 2021, were analyzed for trends and associations within the population. This study used an exploratory approach to analyze three main outcome measures: (i) baseline compliance rate, defined as at least one screening mammogram in the last five years for age eligible women (women 45 years or older, according to ACR recommendations); (ii) no-show rate, defined as women who failed to keep their current mammogram appointments after three attempts at scheduling; and (iii) abnormal mammogram results, defined as high-risk screening outcomes (BI-RADS 4 or 5). Outcomes were specifically analyzed for ABC4WT women between the ages of 40 and 49 years old who would not have been screened according to U.S. Preventative Task Force (USPSTF) guidelines [21]. The American College of Radiology’s risk assessment and quality assurance tool, BI-RADS (Breast Imaging-Reporting and Data System), a widely accepted lexicon and reporting schema for breast imaging, was used for risk differentiation [22]. The minimum sample size requirement was calculated for each main outcome using a 95% confidence interval with a 5% margin for error. The study population was sufficient to detect a statistically significant relationship in all subcategories for the main outcomes, assuming the level of significance alpha (α) = 0.05, apart from two subcategories: “women with BI-RADS 4 or 5” (N2/N1 = 75/1519), using a power of 80% and a margin of error of 7%; and “women who did not know when their last screening took place” (N2/N1 = 103/1371), using a power of 80% and a margin of error less than 6%. Independent variables analyzed for trend analysis included age, race/ethnicity, monthly income, number of persons on income, status of health insurance, county of residence, and results of prior testing if applicable.

Basic descriptive measures (e.g., means, SDs, ranges, histograms, and scatter plots) were obtained to determine the distributions of patient characteristics. SPSS (Statistical Package for the Social Sciences) software version 27 was used to perform transformations, collapse categories, and analyze differences. Differences in patient characteristics were analyzed using appropriate test statistics (*t*-tests for continuous and Pearson’s chi-squared test for categorical-level data). All statistical tests were two-sided, and a *p*-value < 0.05 was considered statistically significant.

Analyses to evaluate the differences in mammogram outcomes and compliance rates by demographics and socioeconomic variables were performed at both the bivariate (Table 1, Table 2, Table 3 and Table 4) and multivariate levels. Covariates considered in the multivariate analyses were age, monthly income, race/ethnicity, and county of residence within COG regions in Texas. Firstly, unadjusted odds ratios (ORs) and 95% confidence intervals (CIs) were reported. A multivariable logistic regression was used to obtain the adjusted odds ratios and the corresponding 95% CIs. To provide a parsimonious model, only those covariates which were significant at the bivariate level were included in the multivriable logistic regression model.

Statement of Ethics and Informed Consent: This study was approved by TTUHSC IRB (Texas Tech University Health Sciences Center, Institutional Review Board). This application was screened for exempt status according to TTUHSC policies and the provisions of applicable federal regulations. This study met criteria for exemption from formal review by the IRB, in accordance with 45 CFR 46.104(d)(4)(iii). Information required to make this determination has been provided by the investigator. All data were in existence at the time of this application for exemption. A waiver of individual HIPAA authorization was requested and found to be appropriate.

## 3. Results

### 3.1. Demographics

The program reached 2065 women from the three Council of Governments (COGs 1, 2, and 7) covered by the ABC4WT project. Characteristics of the study sample organized by outcome variable are presented in Figure 2. There were 1371 (66.4%) participants above the age of 45, and thus expected to have had at least one mammogram within past five years from the time of contact according to the ACR guidelines; however, approximately half of these participants had not been screened in this time (*n* = 576). There were 1788 (86.6%) women eligible for a mammogram at the time of the encounter per the ACR guidelines; 276 (15.4%) of these did not keep the appointments despite three attempts at appointment scheduling (“no-show” rate 15.4%). Of the 1519 women who received mammograms at the time of the contact, 75 women (4.94%) had BI-RADS 4 or 5.

The mean (SD) monthly income and age of ABC4WT participants was USD 654 (665)/month/person and 48.2 (9.8) years old, respectively. Of the 2065 women reached by ABC4WT, 65.4% were Hispanic (*n* = 1351), 28.4% were White (*n* = 587), and 4.5% were Black (*n* = 92). There were 874 (42.3%) participants in COG-1, 981 (47.5%) in COG-2, and 210 (10.2%) in COG-7. Based on the county of residence, participants represented rural areas (*n* = 281 (13.6%)), urban areas (*n* = 1776 (86.0%)), and other (*n* = 7 (0.3%)), as delineated by The Texas Department of Housing and Community Affairs [22]. Over 91% (*n* = 1887) of women included in this study were uninsured at the time of contact.

### 3.2. Baseline Non-Compliance Rate

Program baseline non-compliance (no screening in over five years at program contact) was 58% (*n* = 692); participants in this group represented 52.0% (*n* = 13) of cancer diagnoses and 55.0% (*n* = 22) of negative screening results versus 48.0% (*n* = 12) and 30.0% (*n* = 12) for baseline compliant patients, respectively (*p* = 0.028). Age groups, race, household income per person, and geographical criteria were analyzed as confounding variables (Table 1). Baseline non-compliance was associated with borderline significance to COG region; COG-1 and COG-2 held much higher rates at 51.4% (*n* = 284) and 51.2% (*n* = 352), respectively, than COG-7 at 43.1% (*n* = 56) (*p* = 0.063). Additionally, the baseline non-compliance rate was highest in women who were 40–49 years old (52.7%) and in Black women (52.3%); however, age, race, income, and settlement were not found to be significant factors.

Multivariate analysis demonstrated a borderline significant sociodemographic variable relationship between baseline non-compliance and COG-1 as well as COG-2. The likelihood of participants presenting as baseline non-compliant was increased in COG-1 (AOR = 1.487, 95% CI: 0.977–2.265) and COG-2 (AOR = 1.507, 95% CI: 0.997–2.277). No significant associations with age, settlement, income, insurance, or ethnicity were observed on multivariate analysis.

### 3.3. “No-Show” Rate

There were 1788 eligible for mammograms at the time of participation in ABC4WT program. Despite three attempts at rescheduling, 15.4% (*n* = 276) of these women did not keep their appointments for screening mammograms and were deemed “No-shows” (Table 2). The eligible participants were distributed across all three COGs, with 41.4% (*n* = 740) from COG-1, 49.5% (*n* = 885) from COG-2, and 9.1% (*n* = 163) from COG-7. Patient “no-shows” were significantly more prevalent in COG-1 (24.2%) than COG-2 (10.5%) and COG-7 (2.5%) (*p* < 0.001).

Among participants eligible for screening, Hispanic women were found to have the highest “no-show” rate, at 16.4% compared with Black (9.4%) and White (13.7%) women, albeit this was not statistically significant (*p* = 0.148). Women residing in rural and urban areas had similar “no-show” rates of 15.2% and 15.5%, respectively (*p* = 0.571). Additionally, age and income status were not associated with “no-show” rates.

Multivariate analysis demonstrated significant sociodemographic variable relationships with “no-show” rate: COG-1, COG-2, and age. The likelihood of a participant being a “no-show” was increased in COG-1 (AOR = 11.971, 95% CI: 4.373–32.772) and COG-2 (AOR = 4.428, 95% CI: 1.602–12.238). Women older than 40 years of age were also more likely to no-show appointments than those that were less than 40 years old: ages 40–49 (AOR = 5.697, 95% CI: 2.732–11.878), 50–59 (AOR = 5.185, 95% CI: 2.453–10.959), 60 and older (AOR = 4.446, 95% CI: 1.995–9.911). No significant associations with baseline compliance, insurance, settlement, race, or income were observed on multivariate analysis.

### 3.4. Abnormal Mammograms

A total of 1519 women (73.6%) received ‘no cost’ mammograms through the ABC4WT program; 103 (6.78%) were recommended for biopsy. Of the 103 biopsies performed, 63 (61.2%) resulted in low-risk, 5 (0.2%) in high-risk, and 35 (34.0%) in cancerous lesions. Women between the ages of 40 and 49 years old represented the highest percentages of BI-RADS 4 (42.0%) and BI-RADS 5 (28.0%) when compared with other ages (*p* = 0.049). The cancer rate for this program was 23.04; the national mean is 5.08 per 1000 mammograms. Participants with a monthly household income of less than USD 800/month/person were also more likely to result in a cancer diagnosis following biopsy (70.6%) than higher incomes (29.4%) (*p* = 0.021). 

Factors associated with abnormal mammograms are depicted in Table 3. The mean monthly gross income was not statistically significant when comparing participants that received “normal screening results” (USD 685/month/person) with those that required further diagnostic work-up (USD 659/month/person) (*p* = 0.855). Age was found to be of borderline significance in the outcome of mammogram screening; the mean (±SD) age for normal screening was 49.6 (±8.1) years versus 51.2 (±9.2) years for BI-RADS 4 or 5 (*p* = 0.057). Mammogram outcomes were not associated with income, settlement, race, or region.

Multivariate analysis revealed an increased likelihood of requiring diagnostic procedures if participants resided in COG-2 (AOR = 4.061, 95% CI: 1.599–10.313). Analysis also demonstrated that White women were significantly more likely to require supplemental procedures than Black women (AOR = 2.574, 95% CI: 0.944–17.165). Additionally, no significant difference in procedural requirements was found with Hispanic women or “other” ethnicities, insurance, settlement, age, or income on multivariate analysis.

### 3.5. Women Who Would Not Have Been Screened per USPSTF (U.S. Preventative Task Force) Guidelines (Ages 40–49)

There were 743 (48.9%) women who received ‘no cost’ mammograms through the ABC4WT program aged between ages 40 and 49 years; 43 (5.8%) women were referred for further diagnostic testing. Women between the ages of 40 and 49 years represented the highest percentages of BI-RADS 4 (42.0%) and BI-RADS 5 (28.0%) than other age groups (*p* = 0.040). Participants with a monthly household income of less than USD 800/month/person were also more likely to belong to this age group (48.7%) than higher incomes (37.2%) (*p* < 0.001). 

Factors associated with this subgroup of women are depicted in Table 4. Women aged 40 to 49 years old received more biopsies (43.7%) than other age groups and comprised 28.6% (*n* = 10) of cancers diagnosed (*n* = 35) (*p* = 0.031). Additionally, women aged 40–59 years represented 57.1% of all cancers versus 11.4% for <40 years and 31.4% for >60 years (*p* = 0.031).

Multivariate analysis demonstrated significant sociodemographic variable relationships in the 40–49-year-old age group with COG-2, mammogram outcome, insurance status, and income. This age group was only 1.76 times more likely to receive an outcome of BI-RADS 1, 2, or 3 than BI-RADS 4 or 5 (AOR = 1.754, 95% CI: 1.067–2.890). Women in this age group were also less likely to have insurance (AOR = 0.570, 95% CI: 0.338–0.962) and less likely to have an income of greater than USD 800/month/person (AOR = 0.647, 95% CI: 0.514–0.814). Additionally, participants aged between 40 and 49 years old were more likely to reside in COG-2 than other regions (AOR = 1.436, 95% CI: 0.992–2.079). No significant difference was found for race or settlement on multivariate analysis.

## 4. Discussion

This study was designed to identify potential barriers in breast cancer screening among the underserved women of West Texas. The program primarily targets underserved women; therefore, this population represents a known high-risk group for cancer screening in terms of non-compliance. This analysis provides a more in-depth assessment of risk factors within this population and informs community outreach programs for enhanced strategic planning in outreach efforts.

In this largely uninsured population, we found the highest rate of non-compliance among Black women compared with other ethnicities; this finding is consistent with several studies finding that low-income Black women are more likely to be non-compliant with mammography screening due to various factors, such as the belief that they do not need screening, a lack of health insurance, inadequate breast cancer knowledge, and limited provider referral [23,24,25]. In previous studies, younger women have been described as more compliant with screening than older women, which is contrary to our experience in West Texas [26]. The ABC4WT project provides no-cost screening and assistance with transportation, thereby eliminating a couple of known structural barriers [27]. It also attempts to address cultural and educational barriers via awareness campaign and advocacy. However, the variations in our study’s baseline compliance rates by region, settlement, and income suggest that both cultural and structural barriers have limited these participant groups’ breast cancer screenings [28]. Participants living in COG-1 had the highest rates of baseline non-compliance, which can be attributed to a combination of cultural and structural barriers in these areas. Rural settlement is often found to be a barrier for the equitable delivery of healthcare. A University of Washington Rural Health Research Center study reported that despite improvements in mammographic screening over a span of 11 years, rural women remained less likely to receive mammograms [26]; however, participants in rural areas in our study were not statistically more likely to be non-compliant with screening. One potential reason could be that the ABC4WT project developed several contracts with mammogram providers and designed outreach events in collaboration with local, smaller hospitals, minimizing travel distances for the participants.

In previous studies, access factors and structural barriers, such as health insurance, transportation, and cost, were found to be important barriers to breast cancer screening and were found to limit regular screenings [16,25,29,30]. Our results did not demonstrate a higher rate of non-compliance in women with lower household incomes relative to those with larger household incomes. Therefore, removing structural barriers and improving access (i.e., providing no cost screenings) is likely already effective and has reduced income disparities in screening.

Among the participants eligible for screening, Hispanic women demonstrated the highest “no-show” rate compared with White and Black women. A comparison for these results is difficult to ascertain, because few studies report mammography “no-show” rates in Hispanic women; however, a recent study by Hensing et al., reported Black women having a higher “no-show” rate than non-Hispanic White women. Interestingly, the “no-show” rate for Black women in this program was the lowest of all ethnicities, thus contradicting the findings of Hensing et al.’s study [31]. Another finding in our study is that COG-1 (Texas Panhandle) had the highest rate of “no-show” when compared with COG-2 (South Plains Texas) and COG-7 (Central West Texas). The percentage of “no-show” participants residing in rural areas was unexpectedly similar to the percentage of “no-show” participants in urban settlements (15.2% and 15.5%, respectively). One limitation of our study is that no data on education level, awareness of guidelines, or cultural beliefs were collected, which makes it is difficult to ascertain whether these barriers played a role in missing screening appointments [31,32].

The Breast Cancer Surveillance Consortium reported a national mean cancer rate of 5.1 per 1000 mammograms from the years 2007 to 2013 [33]; the mean cancer rate for this program was 23.04 per 1000 mammograms. These rates demonstrate that despite programs such as ABC4WT, leading to an increase in cancer screening in underserved populations, there are still disparities in both the incidence and mortality of breast cancer. These disparities cannot be fully explained by race, lower economic status (SES), or insurance status [34]. Several studies have demonstrated that women in low-resource settings experience a combination of factors that contribute to lower rates of breast cancer detection and appropriate treatment, as well as poorer survival [35,36,37,38]. ABC4WT addresses the structural barriers of cost, transportation, and “work excuses”; therefore, it is plausible that this discrepancy results from cultural practices of the region, behavioral risk factors, or the socioeconomic status of eligible participants. This insight is valuable to developing strategies and direct resources more focused on overcoming the non-structural cultural barriers in this region.

An increased likelihood of requiring diagnostic procedures was noted if participants resided in COG-2 (South Plains Texas). Interestingly, COG-2 is also the region with the lowest baseline compliance rate. We did not collect data on behavioral risk factors; thus, it is difficult to determine whether the low rate of baseline compliance is the reason for higher rate of abnormal results or if other behavioral factors prevail in the region that might be associated with increased rates of breast cancer.

The mean age for normal screening and abnormal screening results were similar in this analysis at 49.6 years and 51.1 years, respectively. Thus, approximately half of the women with abnormal results would not have received screening per USPSTF guidelines. With women aged between 40 and 49 years old comprising approximately 40% of BI-RADS 4 and 5 and 30% of cancers, this age group represents a high-risk category of participants in the study. Women in this age group were also more likely to be of Hispanic race, have lower incomes, and reside in COG-2, thus making an already at-risk population even more at risk for worse outcomes.

Strengths and limitations of the study: The strengths of this study lie in the extended coverage of three different COG regions encompassing 60 counties. However, not all counties in the target regions were reached by the program. The study was limited in the inferences made with regard to COG-7, which constituted the smallest sampled region in terms of population compared with other regions. A better inference could be derived with an extended study period within this region. Additionally, no data on educational level, guideline awareness, behavioral risk factors, or family history of breast cancer (in a first degree relative) were obtained through this program. Therefore, the association of compliance or “no-show” rate with these factors could not be analyzed. The lack of family history information also makes it difficult to determine if the higher rate of biopsies for women aged 40 to 49 years old is due to a larger proportion of familial history of breast cancer (in a first-degree relative) than other age groups. If indeed so, this may introduce some bias into the analysis.

## 5. Conclusions

Among the underserved population of West Texas, COG region, age, and biopsy outcome were related to a previous lack of compliance with screening. Additionally, despite addressing the income and transportation barriers, a significant “no-show” rate persisted. This is most likely due to cultural and other nonstructural barriers. An understanding of these sociodemographic determinants can enable healthcare providers and public health workers to develop innovative strategies to increase breast cancer screening. The sociodemographic determinants most starkly impacted 40–49-year-old women who would not have been screened by USPSTF guidelines. This population was found to be at increased cancer risk and should be encouraged screening for breast cancer via mammography starting at age 40.

## Figures and Tables

**Figure 1 medicina-58-01010-f001:**
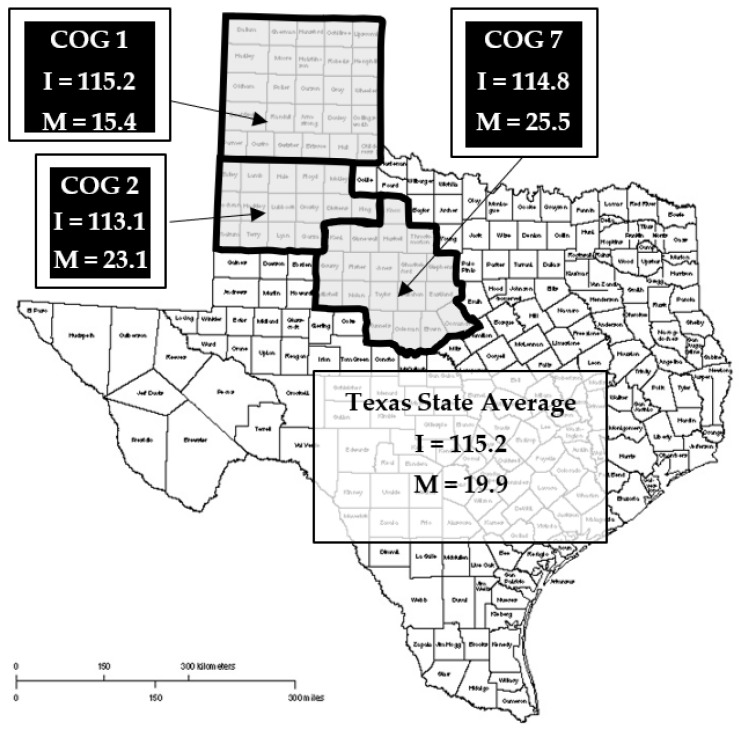
Target region for Access to Breast Care for West Texas (ABC4WT). Outlines mark the Council of Government (COG) regions with age-adjusted incidence (I) and mortality (M) rates for breast cancer per 100,000 women compared with the state averages in 2018.

**Figure 2 medicina-58-01010-f002:**
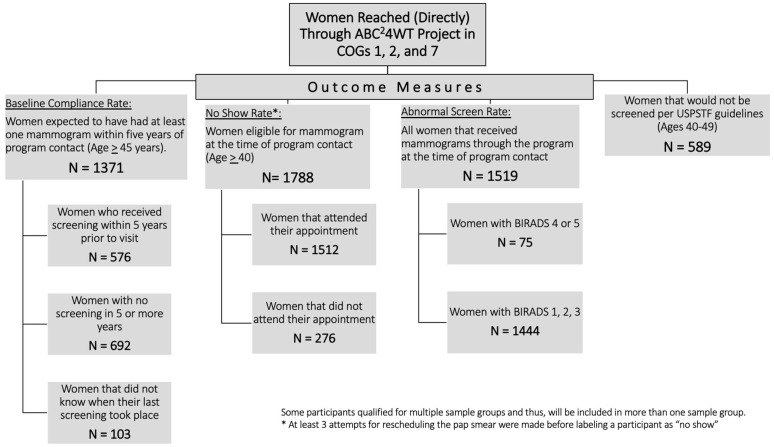
Flow diagram showing the distribution of ABC4WT participants and outcome measures.

**Table 1 medicina-58-01010-t001:** Factors associated with baseline compliance with mammogram screening according to American College of Radiology guidelines.

Characteristics		Mammogram Completed (Baseline Compliance)							*p*-Value
		<5 years		≥5 or Never		Don’t Know		Total Participants in Subgroup, N (%)	
		N	%	N	%	N	%		
**Age** (Years)	**40–49**	**200**	39.8%	265	52.7%	38	7.6%	503 (36.7%)	0.774
	50–59	255	43.3%	289	49.1%	45	7.6%	589 (43.0%)	
	60+	121	43.4%	138	49.5%	20	7.2%	279 (20.4%)	
**Household Income** (USD/month/person)	≤800	350	41.8%	425	50.7%	63	7.5%	838 (64.7%)	0.853
	>800	193	42.1%	227	49.6%	38	8.3%	458 (35.3%)	
**Settlement**	Rural	86	45.5%	94	49.7%	9	4.8%	189 (13.8%)	0.467
	Urban	488	41.5%	595	50.6%	93	7.9%	1176 (85.8%)	
	Other	2	33.3%	3	50.0%	1	16.7%	6 (0.4%)	
**Race**	White	165	41.9%	200	50.8%	29	7.4%	394 (29.2%)	0.991
	Black	17	38.6%	23	52.3%	4	9.1%	44 (3.3%)	
	Hispanic	380	41.8%	462	50.8%	68	7.5%	910 (67.5%)	
**Texas Region**	COG-1	236	42.7%	284	51.4%	33	6.0%	553 (40.3%)	0.063
	COG-2	274	39.8%	352	51.2%	62	9.0%	688 (50.2%)	
	COG-7	66	50.0%	56	43.1%	8	6.2%	130 (9.5%)	
**Biopsy Results**	Non-cancerous	12	30.0%	22	55.0%	6	15.0%	40 (61.5%)	0.028
	Cancerous	12	48.0%	13	52.0%	0	0.0%	25 (38.5%)	

**Table 2 medicina-58-01010-t002:** Factors associated with the “No-Show” rate for mammogram screening appointments.

Characteristics		“No-Show” for Mammogram Appointments			*p*-Value
		N	%	Total Participants in Subgroup, N (%)	
**Age** (Years)	**40–49**	148	16.1%	920 (51.5%)	0.602
	50–59	90	15.3%	589 (32.9%)	
	60+	38	13.6%	279 (15.6%)	
**Household Income** (USD/month/person)	≤800	177	15.5%	1144 (67.4%)	0.293
	>800	75	13.5%	554 (32.6%)	
**Settlement**	Rural	37	15.2%	244 (13.6%)	0.571
	Urban	239	15.5%	1538 (86.0%)	
	Other	0	0.0%	6 (0.3%)	
**Race**	White	67	13.7%	490 (27.4%)	0.148
	Black	6	9.4%	64 (3.6%)	
	Hispanic	198	16.4%	1208 (67.6%)	
**Texas Region**	COG-1	179	24.2%	740 (41.4%)	<0.001
	COG-2	93	10.5%	885 (49.5%)	
	COG-7	4	2.5%	163 (9.1%)	

**Table 3 medicina-58-01010-t003:** Factors associated with “abnormal mammogram results”.

Characteristics		Mammogram Outcomes							*p*-Value
		BI-RADS 1–3		BI-RADS 4		BI-RADS 5		Total Participants in Subgroup, N (%)	
		N	%	N	%	N	%		
**Age** (Years)	**<40**	**72**	91.1%	3	3.8%	4	5.1%	79 (5.2%)	0.049
	40–49	715	96.2%	21	2.8%	7	0.9%	743 (48.9%)	
	50–59	448	95.1%	16	3.4%	7	1.5%	459 (31.0%)	
	60+	209	92.5%	10	4.4%	7	3.1%	218 (14.7%)	
**Household Income** (USD/month/person)	≤800	950	95.2%	30	3.0%	18	1.8%	998 (67.6%)	0.468
	>800	453	94.8%	19	4.0%	6	1.3%	478 (32.4%)	
**Settlement**	Rural	203	95.8%	5	2.4%	4	1.9%	212 (14.0%)	0.897
	Urban	1234	94.9%	45	3.5%	21	1.6%	1300 (85.6%)	
	Other	6	100.0%	0	0.0%	0	0.0%	6 (0.4%)	
**Race**	White	401	94.4%	13	3.1%	11	2.6%	425 (28.4%)	0.366
	Black	56	94.8%	3	5.2%	0	0.0%	59 (3.9%)	
	Hispanic	965	95.4%	33	3.3%	14	1.4%	1012 (67.6%)	
**Texas Region**	COG-1	526	96.2%	16	2.9%	5	0.9%	547 (36.0%)	0.392
	COG-2	761	94.3%	30	3.7%	16	2.0%	807 (53.1%)	
	COG-7	157	95.2%	4	2.4%	4	2.4%	165 (10.9%)	

**Table 4 medicina-58-01010-t004:** Factors associated with mammogram screening, grouped by ages 40–49 years.

Characteristics		Women Eligible per ACS Guideline AND Ineligible per USPSTF (Women Aged 40–49 Years)			*p*-Value
		N	%	Total Participants in Subgroup, N (%)	
**Mammography** Outcome	**BI-RADS 1, 2, 3**	715	49.5%	1443 (95.1%)	0.040
	BI-RADS 4 or 5	28	37.3%	75 (4.9%)	
**Biopsy Result**	Non-cancerous	35	51.5%	68 (66.0%)	0.026
	Cancer	10	28.6%	35 (34.0%)	
**Household Income** (USD/month/person)	≤800	644	48.7%	1323 (68.0%)	<0.001
	>800	232	37.2%	623 (32.0%)	
**Settlement**	Rural	136	48.4%	281 (13.6%)	0.272
	Urban	782	44.0%	1776 (86.0%)	
	Other	2	28.6%	7 (0.3%)	
**Race**	White	213	36.3%	587 (28.9%)	<0.001
	Black	35	38.0%	92 (4.5%)	
	Hispanic	661	48.9%	1351 (66.6%)	
**Texas Region ***	COG-1	363	41.5%	874 (42.3%)	<0.001
	COG-2	493	50.3%	981 (47.5%)	
	COG-7	64	30.5%	210 (10.2%)	

***** COG = Council of Government.

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
