# Peer review of "Sociodemographic Determinants in Breast Cancer Screening among Uninsured Women of West Texas"

_medicina, 2022, doi:10.3390/medicina58081010_

Round 1

Reviewer 1 Report

The manuscript titled "Sociodemographic disparities in breast cancer screening among uninsured women of West Texas" by Jensen et al. makes use of the dataset of the ABC4WT, a public-private-community partnership aimed to identify  barriers and outcomes of breast cancer screening among West Texas underserved women. The study describes general compliance, "no show",  as well as mammography outcomes in women (by age, race, household income, settlement and Texas region) in 2018 through 2021. The screened women were aged 40+y or 10y younger than the age of a first-degree relative diagnosed with breast cancer (unlike the USPSTF recommendations). The results indicated that the age group of 40-49 - which is not covered by the USPSTF - represented the largest percentage of BIRADS-4 and BIRADS-5 findings, and had a high incidence rate of breast cancer.

GENERAL COMMENTS

The study question is of relevance and the design and methods are sound. However, in a study based on an existing dataset, more information should be presented regarding the completeness and validity of the data analyzed. In addition, the study mostly addresses women aged 40+y, but includes both women at an average risk for breast cancer and women at a higher risk due to having a first-degree relative who was diagnosed with breast cancer. It is likely that having a mother/daughter/sister with breast cancer affects attitudes and practices of screening, but the authors did not stratify the study population by this variable, nor have they discussed it as a limitation in the Discussion.

SPECIFIC COMMENTS

1. INTRODUCTION - It would be helpful to add some information about sociodemographic differences between the three COGs studied (1,2,7).

2. METHODS - Please expand the definitions of the outcome variables. For the non-compliance variable, were also women who performed no mammogram at all included? how far back, time-wise, is the dataset covering? was this analysis restricted for 40+y women only? 45+y women only (to allow for a five past years)? 

For the abnormal mammogram results, was BIRADS-2,3 considered low risk, and BIRADS-4,5, high risk? how was breast cancer defined? following a biopsy?

3. METODS - was information on first-degree relatives with breast cancer available? if so, it should have been taken into consideration, and analyses should have been stratified by this variable.

4. RESULTS - Figure 2 is missing. 

5. RESULTS - The tables are presented in an extremely small and unclear font.

6. RESULTS - Table 1 - the headline immediately above the table reads "No-show rate" and seems to be misplaced.

7. RESULTS - Bottom of page 5 - Women less than 40y, are - according to the authors' description - always first degree relatives of breast cancer patients, who started their screening at an age which is 10y younger than their relative's diagnosis age. Thus, it makes sense that these women, who are at a higher-than-average risk for the disease, will have a lower "no show" rate.

Similarly, at the bottom of page 7 it is said that women aged 40-49 received more biopsies than other age groups. It is crucial to understand whether this age group included a larger proportion of familial history of breast cancer (in a first degree relative) than other age groups. If indeed so, this may explain the results, at least partially.

This point should be addressed in the analyses as well as the Discussion.

8. RESULTS - Are the tables presenting univariable analyses? multivariable? the headlines are not specifying this. I had the impression that the analyses in the tables are univariable, and would like to suggest that tabulated multivariable results may be added. In addition, were "compliance" a covariate in the adjusted analyses? 

9. DISCUSSION - mid page 11 - it should be clarified if the national rate of breast cancer per 1,000 mammograms is based on the USPSTF guidelines (average risk women aged 50-74y) or on the guidelines implemented in this study. If the former, this may explain, at least partially, the differences observed.

10. DISCUSSION - another limitation is that the data were not analyzed by level of risk.  I would advise repeating the analyses taking this point into account, if possible. If not, this potential source for both selection and information bias should be discussed.

Author Response

  1. INTRODUCTION - It would be helpful to add some information about sociodemographic differences between the three COGs studied (1,2,7).
    1. Response: Information about the sociodemographic differences in the three COGs was added in the introduction.
  2. METHODS - Please expand the definitions of the outcome variables. For the non-compliance variable, were also women who performed no mammogram at all included? how far back, time-wise, is the dataset covering? was this analysis restricted to 40+y women only? 45+y women only (to allow for five past years)? 
    1. Response: The non-compliance variable and the no-show variable were not mutually exclusive. There is a subgroup of patients that were both non-compliant and "no-shows". These women were included in the sociodemographic categories for non-compliance but were not included in the outcomes section. Thus, the overall "n" in each category is important. More clarification was added for the outcome variables in the methods section to help clarify this issue.
  3. METHODS- For the abnormal mammogram results, was BIRADS-2,3 considered low risk, and BIRADS-4,5, high risk? how was breast cancer defined? following a biopsy?
    1. Response: Breast cancer was defined as a diagnosis resulting from a biopsy demonstrating cancerous cells. Pre-cancerous cells on biopsy were not included in the "breast cancer" diagnosis. "Low risk" was determined via biopsy as a pre-cancerous lesion with a lower likelihood per Radiologist of transforming into a cancerous lesion. "High risk" was also determined via biopsy as a pre-cancerous lesion with a higher likelihood per Radiologist of transforming into a cancerous lesion.
  4. METHODS - was information on first-degree relatives with breast cancer available? if so, it should have been taken into consideration, and analyses should have been stratified by this variable.
    1. Response: The participants' family history was, unfortunately, unavailable for analysis.
  5. RESULTS - Figure 2 is missing. 
    1. Response: Figure 2 has been added to the manuscript.
  6. RESULTS - The tables are presented in an extremely small and unclear font.
    1. Response: Font has been adjusted for better clarity.
  7. RESULTS - Table 1 - the headline immediately above the table reads "No-show rate" and seems to be misplaced.
    1. Response: Adjusted Table 1's position in the manuscript.
  8. RESULTS - Bottom of page 5 - Women less than 40y, are - according to the authors' description - always first-degree relatives of breast cancer patients, who started their screening at an age which is 10y younger than their relative's diagnosis age. Thus, it makes sense that these women, who are at a higher-than-average risk for the disease, will have a lower "no show" rate.
    1. Response: That is absolutely correct and we regret not having considered that in our initial analysis. We have added that discussion point to the manuscript.
  9. Similarly, at the bottom of page 7, it is said that women aged 40-49 received more biopsies than other age groups. It is crucial to understand whether this age group included a larger proportion of familial history of breast cancer (in a first-degree relative) than other age groups. If indeed so, this may explain the results, at least partially. This point should be addressed in the analyses as well as the Discussion.
    1. Response: Unfortunately family history was unavailable but that will be added to our list of limitations in this study.
  10. RESULTS - Are the tables presenting univariable analyses? multivariable? the headlines are not specifying this. I had the impression that the analyses in the tables are univariable, and would like to suggest that tabulated multivariable results may be added. In addition, was "compliance" a covariate in the adjusted analyses? 
    1. Response: The tables presented are the univariable analyses - the multivariable analyses are explained in the manuscript but not presented in a visual format. Multivariate analyses were not added due to the fluctuating number of participants eligible for each category. We felt the descriptive statistics would be a better representation of the studied population characteristics. 
  11. DISCUSSION - mid-page 11, it should be clarified if the national rate of breast cancer per 1,000 mammograms is based on the USPSTF guidelines (average-risk women aged 50-74y) or on the guidelines implemented in this study. If the former, this may explain, at least partially, the differences observed.
    1. Response: This has been clarified in the manuscript.
  12. DISCUSSION - another limitation is that the data were not analyzed by the level of risk.  I would advise repeating the analyses taking this point into account, if possible. If not, this potential source for both selection and information bias should be discussed.
    1. Response: The potential bias has been added to the limitations section.

Reviewer 2 Report

Jensen et al discussed the impact of socioeconomic and sociodemographic factors in the non-compliance of underserved and/or uninsured women treated by ABC4WT in three West Texas regions, Council of Government Region-1, -2, and -7. The objective of this study is to identify the disparities that exist in West Texas' low-income, uninsured, or under-insured communities, inferring that the economic burden of breast cancer in these regions (COG-1, COG-2, and COG-7) is notable due to the higher incidence rates than the state's average. This analysis provides a more comprehensive assessment of risk factors in this population and suggests community outreach programs for improved strategic planning in outreach efforts. The manuscript is well-designed and presented, and it elucidates key risk factors associated with breast cancer incidence. However, as the authors have mentioned, data on educational level, guideline awareness, and behavioral risk factors appear to be important and could be used to supplement this study.

Some issues require the author's attention:

1.   Figure 1: A better representative image is required because the current one lacks clarity regarding COG-2 regions.

2.   The manuscript contains numerous acronyms, and a list of acronyms will aid in comprehension. Some acronyms, such as 'BIRADS' and 'MQSA,' should be defined.

3.  Line 155/156: Figure 2 is missing from the manuscript and needs to be added.

Author Response

  1. Figure 1: A better representative image is required because the current one lacks clarity regarding COG-2 regions.
    1. Response: The image has been adjusted for better clarity and comprehension.
  2. The manuscript contains numerous acronyms, and a list of acronyms will aid in comprehension. Some acronyms, such as 'BIRADS' and 'MQSA,' should be defined.
    1. Response: A list of used acronyms has been added to the manuscript underneath the abstract (lines 32-36)
  3. Line 155/156: Figure 2 is missing from the manuscript and needs to be added.
    1. Response: Figure 2 has been added to the manuscript.

Reviewer 3 Report

The authors aimed to assess the role of socioeconomic and sociodemographic factors in the non-compliance of underserved/uninsured women treated by ABC4WT in three regions of West Texas. I would like to congratulate the authors for their effort. However, I consider that there are several aspects that should be considered/improved:

Major comments:

1. The authors use the term: "Sociodemographic Disparities". However, after reading the manuscript, it seems to me that the most appropriate term would be "Sociodemographic Determinants". I think that, in order to assess "disparities", according to an epidemiological and statistical perspective, the correct approach would have been presenting the results stratified (eg. assessing the sociodemographic variable as a potential effect modifier) in a context of an epidemiological approach (X --> Y, adjusted by confounders and stratified by sociodemographics in order to see if there are "disparities" [ie. variations across categories of the variable] in the association between the exposure and the outcome).

2. The authors mention that "The study population was sufficient to detect a statistically significant relationship assuming the level of significance, alpha (α) = 0.05". Relationship between which variables? It is important to mention that this study used an exploratory approach (eg. "Factors associated"). For this reason, it is important to mention which were the proportions used for the sample size calculation (and for which variables were calculated), at what level the statistical power was set (eg. 80%), what was the N2/N1 ratio, etc. An alternative could be the calculation of a post-hoc power calculation, in order to be sure that we are not facing a type 2 error. In fact, in some cases I see wide confidence intervals, which might be a sign that power is lacking.

3. It is not clear why the authors opted for the estimation of odds ratio instead of prevalence ratios (eg. estimated by a GLM log-Poisson with robust variances), since some outcomes have a % greater than 10%, considering the potential overestimation of using OR with this high %.

4. It is not clear how the participants in the "Don't know" group were handled. Were they excluded? Were they included in the reference category? It is important to be sure how this was handled as it could be a source of bias in the analysis.

5. It is not clear why the authors presented in tables the descriptive statistics instead of the regression models (crude and adjusted), since I understand this would be the manner of answering their research question. Consider doing this (descriptive statistics could be presented as supplementary).

Minor comments:

1. I can’t find the Figure 2 in the manuscript

Author Response

  1. The authors use the term: "Sociodemographic Disparities". However, after reading the manuscript, it seems to me that the most appropriate term would be "Sociodemographic Determinants". I think that, in order to assess "disparities", according to an epidemiological and statistical perspective, the correct approach would have been presenting the results stratified (eg. assessing the sociodemographic variable as a potential effect modifier) in a context of an epidemiological approach (X --> Y, adjusted by confounders and stratified by sociodemographics in order to see if there are "disparities" [ie. variations across categories of the variable] in the association between the exposure and the outcome).
    1. Response: The manuscript and its title have been adjusted to reflect this feedback.
  2. The authors mention that "The study population was sufficient to detect a statistically significant relationship assuming the level of significance, alpha (α) = 0.05". Relationship between which variables? It is important to mention that this study used an exploratory approach (eg. "Factors associated"). For this reason, it is important to mention which were the proportions used for the sample size calculation (and for which variables were calculated), at what level the statistical power was set (eg. 80%), what was the N2/N1 ratio, etc. An alternative could be the calculation of a post-hoc power calculation, in order to be sure that we are not facing a type 2 error. In fact, in some cases, I see wide confidence intervals, which might be a sign that power is lacking.
    1. Response: The methods section has been amended and the requested values have been added to the manuscript.
  3. It is not clear why the authors opted for the estimation of odds ratio instead of prevalence ratios (eg. estimated by a GLM log-Poisson with robust variances), since some outcomes have a % greater than 10%, considering the potential overestimation of using OR with this high %.
    1. Response: The adjusted odds ratio was used rather than the prevalence ratios due to the statistical database software that was used to perform the multivariate analysis. The software calculated the AOR while computing the multivariate analysis and thus we felt that it would be the most valuable statistic to readers since it took into account all relevant variables. We will, however, recalculate the prevalence ratios if you would like.
  4. It is not clear how the participants in the "Don't know" group were handled. Were they excluded? Were they included in the reference category? It is important to be sure how this was handled as it could be a source of bias in the analysis.
    1. Response: The participants in the “Don’t Know” group were not excluded and were listed as their own category; however, they did have a lower statistical power due to the smaller sample size.
  5. It is not clear why the authors presented in tables the descriptive statistics instead of the regression models (crude and adjusted), since I understand this would be the manner of answering their research question. Consider doing this (descriptive statistics could be presented as supplementary).
    1. Response: 

      The tables presented are the univariable analyses - the multivariable analyses are explained in the manuscript but not presented in a visual format. Multivariate analyses were not added visually due to the fluctuating number of participants eligible for each category. We felt the descriptive statistics would be a better representation of the studied population characteristics and less confusing to readers.

Minor comments:

  1. I can’t find the Figure 2 in the manuscript
    1. Response: Figure 2 has been added to the manuscript.

Round 2

Reviewer 1 Report

Re. point 10:

RESULTS - Are the tables presenting univariable analyses? multivariable? the headlines are not specifying this. I had the impression that the analyses in the tables are univariable, and would like to suggest that tabulated multivariable results may be added. In addition, was "compliance" a covariate in the adjusted analyses? 

The authors did not refer to the point of adding "compliance" to the covariates. 

Author Response

RESULTS - Are the tables presenting univariable analyses? multivariable? the headlines are not specifying this. I had the impression that the analyses in the tables are univariable, and would like to suggest that tabulated multivariable results may be added.

The tables presented are bivariate analyses. Multivariable analyses were performed but not present in table format. They are described in the last paragraph of each Result subsection.

In addition, was "compliance" a covariate in the adjusted analyses? The authors did not refer to the point of adding "compliance" to the covariates. 

Baseline compliance was not a covariate in the adjusted analyses; however, "no show" was a covariate included in the multivariable analyses.